



# N$_2$O as a regression proxy for dynamical variability in stratospheric trace gas trends

Kimberlee Dubé[1], Susann Tegtmeier[1], Adam Bourassa[1], Daniel Zawada[1], Doug Degenstein[1], Patrick E. Sheese[2], Kaley A. Walker[2], and William Randel[3]

[1]Institute of Space and Atmospheric Studies, University of Saskatchewan, Saskatoon, SK, Canada
[2]Department of Physics, University of Toronto, Toronto, ON, Canada
[3]National Center for Atmospheric Research, Boulder, CO, USA

**Correspondence:** Kimberlee Dubé (kimberlee.dube@usask.ca)

**Abstract.** Trends in stratospheric trace gases like HCl, N$_2$O, O$_3$, and NO$_y$ show a hemispheric asymmetry over the last two decades, with trends having opposing signs in the Northern and Southern Hemispheres. Here we use N$_2$O, a long-lived tracer with a tropospheric source, as a proxy for stratospheric circulation in the multiple linear regression model used to calculate stratospheric trace gas trends. This is done in an effort to isolate trends due to circulation changes from trends due to ozone depleting substances. We use measurements from the Atmospheric Chemistry Experiment Fourier Transform Spectrometer (ACE-FTS) and the Optical Spectrograph and InfraRed Imager System (OSIRIS), and model results from the Whole Atmosphere Community Climate Model (WACCM). Trends in HCl, O$_3$, and NO$_y$ for 2004–2018 are examined. Using the N$_2$O regression proxy, we show that observed HCl increases in the Northern Hemisphere are due to changes in the stratospheric circulation. We also show that negative O$_3$ trends above 30 hPa in the Northern Hemisphere can be explained by change in the circulation, but that negative ozone trends at lower levels cannot. Trends in stratospheric NO$_y$ are found to be largely consistent with trends in N$_2$O.

## 1 Introduction

The stratospheric ozone layer is critical for the existence of life on Earth, as it absorbs harmful solar ultraviolet (UV) radiation. The discovery of declining ozone concentrations in the final few decades of the 20th century was thus of great concern. Stratospheric ozone loss was attributed to elevated levels of halogen-containing Ozone Depleting Substances (ODSs) (Solomon, 1999). The Montreal Protocol and its amendments successfully reduced anthropogenic emissions of ODSs (Laube et al., 2022), and recent observations show that upper stratospheric (∼32-50 km) ozone has increased at a rate of 1%–3% per decade since the beginning of the 21st century (e.g. Steinbrecht et al., 2017; Ball et al., 2018; Bourassa et al., 2018; SPARC/IO3C/GAW, 2019; Bognar et al., 2022; Godin-Beekmann et al., 2022). However, ozone trends in the lower stratosphere (below ∼24 km), and particularly in the Northern Hemisphere (NH) are insignificant, or even negative, over the same time period (e.g. Ball et al., 2018, 2019; Wargan et al., 2018; SPARC/IO3C/GAW, 2019; Bognar et al., 2022). The exact cause of the observed lower stratospheric (LS) ozone trends is uncertain: Abalos et al. (2021) and Orbe et al. (2020) found the LS ozone trends to be caused by changes in the Brewer-Dobson circulation (BDC), while Oberländer-Hayn et al. (2016) and Match and Gerber





(2022) associated the LS ozone decrease with an increase in the tropopause height. The increased tropopause height is caused

by rising greenhouse gas (GHG) emissions that are warming the troposphere and cooling the stratosphere (Vallis et al., 2015). The mechanism behind the hemispherically asymmetric BDC changes is some combination of GHG emissions, ozone recovery, and internal variability (e.g. Abalos et al.; Strahan et al., 2020; Ploeger and Garny, 2022). Trends in hydrogen chloride (HCl, Mahieu et al., 2014; Strahan et al., 2020), nitric acid ($HNO_3$, Strahan et al., 2020), reactive fluorine ($F_y$, Prignon et al., 2021), nitrous oxide ($N_2O$, Ploeger and Garny, 2022; Minganti et al., 2022), and nitrogen oxides ($NO_x=NO_2+NO$, Yela

et al., 2017; Galytska et al., 2019; Dubé et al., 2020) each show a similar trend pattern to $O_3$, with opposing signs in the lower stratosphere NH relative to the Southern Hemisphere (SH) .

   Stratospheric trace gas trends are typically calculated using either a Multiple Linear Regression (MLR) model (e.g. Bourassa et al., 2018; SPARC/IO3C/GAW, 2019), or a Dynamic Linear Model (DLM) (e.g. Ball et al., 2019; Bognar et al., 2022). In both cases it is necessary to represent phenomena that are known to affect trace gas concentrations by proxy variables. This is

inherently difficult as we generally do not have the information needed to represent sources of variability in a regression model accurately. In particular, it is common to only consider dynamical changes caused by the Quasi-Biennial Oscillation (QBO) and the El-Niño Southern Oscillation (ENSO). Changes in the BDC are typically neglected due to the lack of an observation-based proxy, even though models have shown that they have an impact on ozone (e.g. Wargan et al., 2018; Chipperfield et al., 2018). Neglecting to consider the BDC in regression models makes it difficult to attribute the cause of the derived trends:

the trends contain influences from both changing chemistry and changing dynamics. If trends in stratospheric circulation are correctly accounted for in the regression model, the remaining trends can be attributed to changes in chemistry (or some other unknown/unaccounted for process). Knowledge of the ozone trend due to chemistry is of particular importance for assessing the impact of the Montreal Protocol on preventing further ozone destruction by ODSs.

   Here we consider using $N_2O$ as a proxy for dynamic variability in the MLR model used to calculate stratospheric trace

gas trends. We refer to a MLR model containing only a linear trend, a constant, and an $N_2O$ proxy as the "$N_2O$ MLR", and the more typical regression model with dynamical variability represented by QBO and ENSO proxies as the "standard MLR" (see Section 3 for detailed definitions). $N_2O$ is a long-lived tracer with a tropospheric source and a known surface trend so it provides a good representation of transport anomalies throughout the stratosphere. This method was originally proposed by Stolarski et al. (2018), who used it to determine stratospheric HCl trends based on observations from the Microwave Limb

Sounder (MLS, Waters et al., 2006). Mahieu et al. (2014) had previously observed an increase in stratospheric HCl in the NH, beginning around 2007, despite decreasing levels of chlorine-containing ODSs (Laube et al., 2022). Mahieu et al. (2014) attributed the elevated NH HCl to a slowing-down of the NH branch of the BDC. Stolarski et al. (2018) further verified this theory by showing that HCl trends calculated using an $N_2O$ MLR are negative from 45°N-50°N. This implies that the positive NH HCl trends obtained from a simple linear regression, without an $N_2O$ proxy (Mahieu et al., 2014; Stolarski et al., 2018),

as well as with the standard MLR (Froidevaux et al., 2019), are due to changes in circulation rather than changes in chlorine emissions.

   Recent studies have used the $N_2O$ MLR method to determine trends in other gases and from other instruments. Bernath and Fernando (2018) determined the global average HCl trend in observations from the Atmospheric Chemistry Experiment -



Fourier Transform Spectrometer (ACE-FTS, Bernath et al., 2005) to be -5 %/decade from 2004 to 2017, which is in agreement
with the trend in tropospheric chlorine. Zambri et al. (2019) used the $N_2O$ MLR to remove dynamical variability from $NO_y$
observations in order to isolate the influence of volcanic aerosol, while Hannigan et al. (2022) determined trends in carbonyl
sulfide (OCS) using the $N_2O$ MLR.

Here we provide an update to the $N_2O$ MLR approach presented in Stolarski et al. (2018). Results are shown from 2004–
2018; for simplicity we call linear changes over this 15 year time period "trends", although trends are generally understood
to occur on multi-decadal scales. We focus on observations from ACE-FTS in order to avoid difficulties caused by the known
drift in the MLS $N_2O$ observations (Livesey et al., 2021). HCl trends are provided in 10° latitude and 1 km altitude bins, rather
than only from 45°N-50°N (Stolarski et al., 2018) or the global average (Bernath and Fernando, 2018). We also provide ozone
trends calculated with the $N_2O$ MLR for the first time. While earlier studies have included a proxy for the BDC in the ozone
regression, they focused on column ozone and used the eddy heat flux (EHF) at 100 hPa from a reanalysis as the proxy, instead
of $N_2O$ observations. There results were also inconsistent: SPARC/IO3C/GAW (2019) found the inclusion of an EHF proxy
to have a negligible effect on ozone trends, while Weber et al. (2022) showed that including the EHF proxy resulted in column
ozone trends that agreed better with the expected values based on ODSs than the ozone trends calculated with the standard
MLR. In addition to HCl and $O_3$, we also consider the relationship between $N_2O$ trends and $NO_y$ trends. Trends in both
free-running and specified dynamics simulations from the Whole Atmosphere Community Climate Model (WACCM), and in
observations from the Optical Spectrograph and InfraRed Imager System (OSIRIS, Llewellyn et al., 2004) are investigated
along with trends in ACE-FTS observations, and the feasibility of using both $N_2O$ modelled by WACCM and $N_2O$ measured
by ACE-FTS as the regression proxy is assessed.

## 2    Data

### 2.1    Satellite Observations

ACE-FTS is an infrared Fourier transform spectrometer that measures from 750–4400 cm$^{-1}$ (Boone et al., 2005; Bernath
et al., 2005). ACE-FTS is in a high inclination orbit, using solar occultation viewing geometry to make approximately 30
atmospheric transmission profile measurements each day: ∼15 at sunrise and ∼15 at sunset. Vertical profiles of over 40 trace
gas species are retrieved from ACE-FTS measurements. Here we consider version 4.2 of the retrieval (Boone et al., 2020) for
several molecules. The observations are filtered according to the data quality flags developed by Sheese et al. (2015).

Vertical profiles of ozone number density are retrieved from OSIRIS measurements of limb-scattered sunlight between 280–
800 nm (Llewellyn et al., 2004). OSIRIS measures 100-400 profiles each day. The version 7.2 OSIRIS $O_3$ retrieval, described
in Bognar et al. (2022), is used here. Only the descending node measurements, with a local time near 6:30 AM, are used when
calculating $O_3$ trends. These measurements make up the majority of observations as the precessing orbit has led to the loss of
ascending node measurements over time.

Both the ACE-FTS and OSIRIS profiles are filtered to remove observations with uncertainties greater than 100%. The
pressure and temperature profiles provided with the data are then used to convert the vertical scale of each profile from altitude



to pressure so that the results can be more easily compared with WACCM. For ACE-FTS the temperatures and pressures are retrieved from the instrument's observations, while for OSIRIS they are from the Modern-Era Retrospective analysis for Research and Applications Version 2 (MERRA-2, Gelaro et al., 2017). Lastly, the area weighted monthly zonal mean (MZM)

is calculated in 10 degree latitude bins and months with fewer than 5 observations in a bin are excluded.

The ACE-FTS $NO_y$ is calculated as

$$NO_y = NO + NO_2 + 2N_2O_5 + HNO_3 + ClONO_2. \tag{1}$$

The monthly zonal mean profiles are used, rather than individual profiles. Sunrise and sunset occultations are kept separate in order to avoid the influence of significant diurnal variations in NO and $NO_2$.

## 100   2.2   $N_2O$ surface emissions

A times series of $N_2O$ surface emissions is needed to determine the stratospheric $N_2O$ trend solely due to changes in the BDC. We use global monthly mean $N_2O$ measurements from the National Oceanic and Atmospheric Administration Global Monitoring Laboratory (NOAA/GML) Halocarbons and other Atmospheric Trace Species (HATS) flask sampling program (NOAA/GML, 2022). The combined $N_2O$ dataset incorporates monthly mean measurements from 13 stations and from 1977–

105   2022.

## 2.3   WACCM

The satellite observations from ACE-FTS and OSIRIS are compared to results from WACCM, a coupled chemistry-climate model. The WACCM results used here are from version 6 of the model, described in Gettelman et al. (2019). WACCM6 has 70 vertical levels extending from the surface to 140 km, and a horizontal resolution of $0.95°$ latitude by $1.25°$ longitude.

The free-running WACCM results used here follow the REFD1 scenario, which includes forcing from observed sea surface temperatures, greenhouse gases, ozone depleting substances, and volcanic aerosol (Plummer et al., 2021). The Quasi-Biennial Oscillation (QBO) was nudged to match observations. $N_2O$, HCl, $O_3$, and $NO_y$ from four ensemble members with slightly different initial conditions are considered.

Results from a WACCM6 specified dynamics (SD) run are also used. In WACCM-SD the winds, temperatures, and surface

fields are nudged to match values from MERRA-2, which constrains the dynamical variability in the model (Gettelman et al., 2019). The WACCM-SD run has 88 vertical levels from the surface to 140 km, corresponding to the MERRA-2 levels.

All WACCM results are provided as monthly zonal means with a $0.95°$ latitude resolution. This is downsampled to $10°$ latitude bins before performing the analysis. It is not possible to resample WACCM to match the times and locations of the ACE-FTS and OSIRIS observations accurately because the WACCM results are only available as monthly zonal means.

Interpolating the WACCM values to the latitudes that an instrument measured in a given month before downsampling to the $10°$ latitude had a negligible impact on the resulting trends, so for simplicity none of the WACCM results shown here are resampled to match the observations.



## 3 Method

To calculate the standard trends we use the Long-term Ozone Trends and Uncertainties in the Stratosphere (LOTUS) regression

code (SPARC/IO3C/GAW, 2019). For each latitude and altitude bin the standard MLR equation describing the concentration

of a gas, $y(t)$, is

$$
\begin{aligned}
y(t) = {} & \beta^{(2)} + \beta_{trend} \times linear(t) + \beta_{qboa}^{(2)} \times QBO_a(t) + \beta_{qbob}^{(2)} \times QBO_b(t) \\
& + \beta_{solar} \times F10.7(t) + \beta_{enso} \times ENSO(t) + \beta_{aod} \times AOD(t) + R(t).
\end{aligned}
\tag{2}
$$

Each $\beta$ is a regression coefficient, with the superscripts specifying the number of the highest seasonal harmonic included

for a given term. $QBO_a(t)$ and $QBO_b(t)$ are the first two principal components of the Singapore zonal winds, $F10.7(t)$ is

the solar flux at 10.7 cm, $ENSO(t)$ is the multivariate ENSO index, and $R(t)$ is the residual. These predictors are further

described in SPARC/IO3C/GAW (2019). $AOD(t)$ is the aerosol optical depth at 525 nm, from the Global Space-based Strato-

spheric Aerosol Climatology (version 2.2) (GloSSAC, Kovilakam et al., 2020). All regression proxies are the same in each

latitude/altitude bin, except for $AOD(t)$ which varies with latitude.

Figure 1 shows the $N_2O$ trend in ACE-FTS observations and WACCM results from 2004/02 – 2018/12 calculated using

the standard MLR. It should be noted that the units of %/decade, which are used here to remain consistent when discussing

$O_3$ trends in terms of earlier studies, result in large trend values where the $N_2O$ concentration is low, for example in the

upper stratosphere. The $N_2O$ trends in units of ppbv/decade are shown in the appendix, Figure A1. Overall there is very good

agreement between the trends in $N_2O$ observed by ACE-FTS and the trends in $N_2O$ modelled by WACCM. The trends in the

WACCM specified dynamics run are the most similar to the trends observed by ACE-FTS, particularly in the NH and in the

upper stratosphere. Trends in the free-running WACCM ensemble members are also remarkably similar to trends in ACE-FTS

$N_2O$. In all cases there is a clear hemispheric asymmetry in the trends, with negative values in the NH below 20 hPa, and

positive values in the SH below 20 hPa. At higher levels this pattern switches and there are lower trends in the SH relative to

the NH. This asymmetry in the $N_2O$ trends has been discussed in earlier studies from Galytska et al. (2019), Ploeger and Garny

(2022), and Minganti et al. (2022), who all attributed the trends to changes in the BDC that are causing the air to become older

in the NH lower stratosphere relative to the SH lower stratosphere. It is worth noting that Minganti et al. (2022), who used the

same REFD1 simulations, found it necessary to resample WACCM at ACE-FTS measurement locations for the hemispheric

asymmetry below 20 hPa to appear. This is not what we observe here, where the overall structure of the WACCM and ACE-FTS

trends are very similar without any resampling of WACCM.

The stratospheric $N_2O$ trends depend not only on circulation changes, but also on changing $N_2O$ surface emissions. This

emission trend needs to be removed from the ACE-FTS $N_2O$ data and the WACCM $N_2O$ results before the $N_2O$ can be used

as a proxy for dynamic variability. The global surface trend in $N_2O$ abundance measured by the NOAA/GML HATS flask

program is calculated using a simple linear regression,

$$
y(t) = \beta + \beta_{trend} \times linear(t) + R(t).
\tag{3}
$$

The surface $N_2O$ data and the trend lines are shown in the top panel of Figure 2. The trend is 2.8 %/decade for 2004–2018.



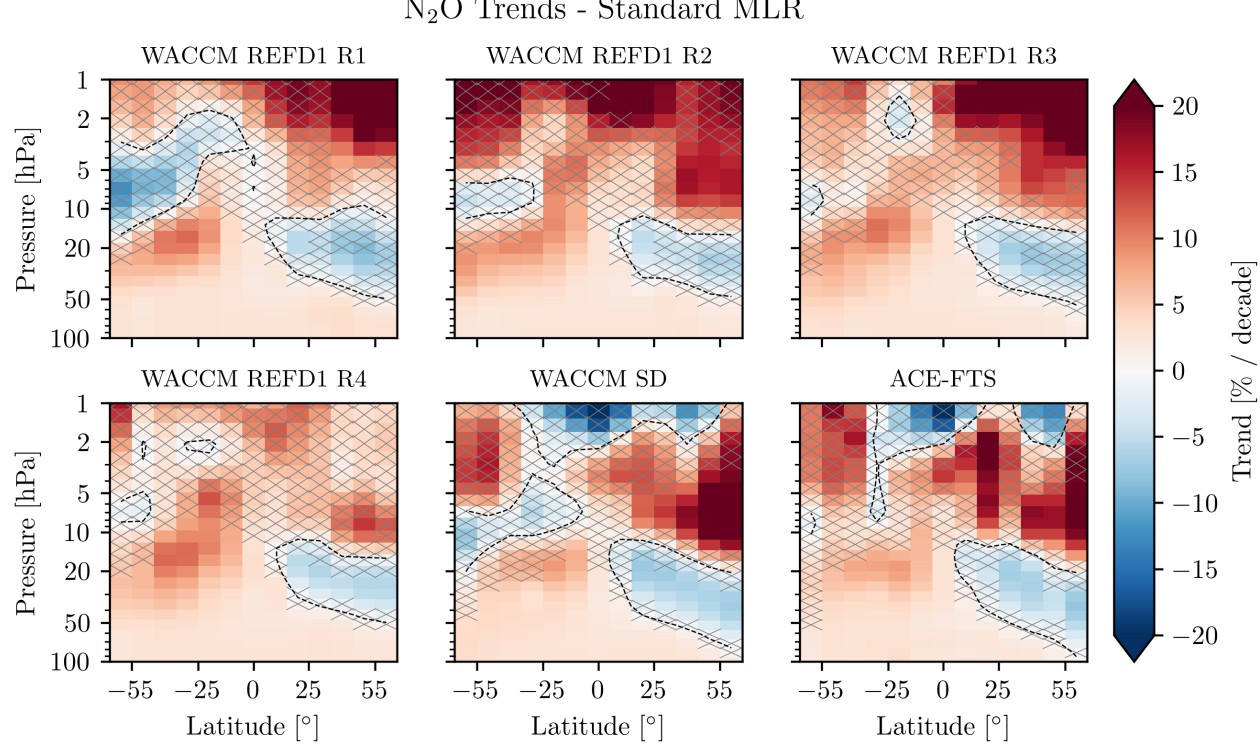

**Figure 1.** $N_2O$ trend in ACE-FTS and WACCM for 2004/02 – 2018/12, as calculated with the standard MLR (Equation 2). R1 to R4 are different ensemble members from the free-running WACCM. Hatched regions are insignificant at the $2\sigma$ level. Dashed contours mark the transitions from positive to negative trends.

To construct the $N_2O$ regression proxy, the trend in the surface $N_2O$ anomaly is subtracted from each latitude and altitude bin of the MZM stratospheric $N_2O$ anomalies. By working with the anomaly we account for differences in the absolute $N_2O$ concentrations between the surface $N_2O$ measurements and the ACE-FTS measurements or WACCM results. The bottom row of Figure 2 shows the portion of the $N_2O$ trend that remains once the increasing surface $N_2O$ emissions are accounted for.
Only one free-running WACCM ensemble member is included as an example. We are assuming that this remaining $N_2O$ trend is largely due to circulation changes in the atmosphere. There could also be changes in the photolysis rate of $N_2O$ and the rate of reaction with $O(^1D)$, however Prather et al. (2023) estimated this effect to be minimal compared to the total $N_2O$ trend: about 1%/decade in the region of maximum photolysis (30°S–30°N, 3–15 hPa).

The MZM $N_2O$ time series with the surface trend removed is used as a proxy in the $N_2O$ MLR equation defined by Stolarski
et al. (2018),

$$y(t) = \beta + \beta_{trend} \times linear(t) + \beta_{n2o} \times N_2O(t) + R(t). \tag{4}$$



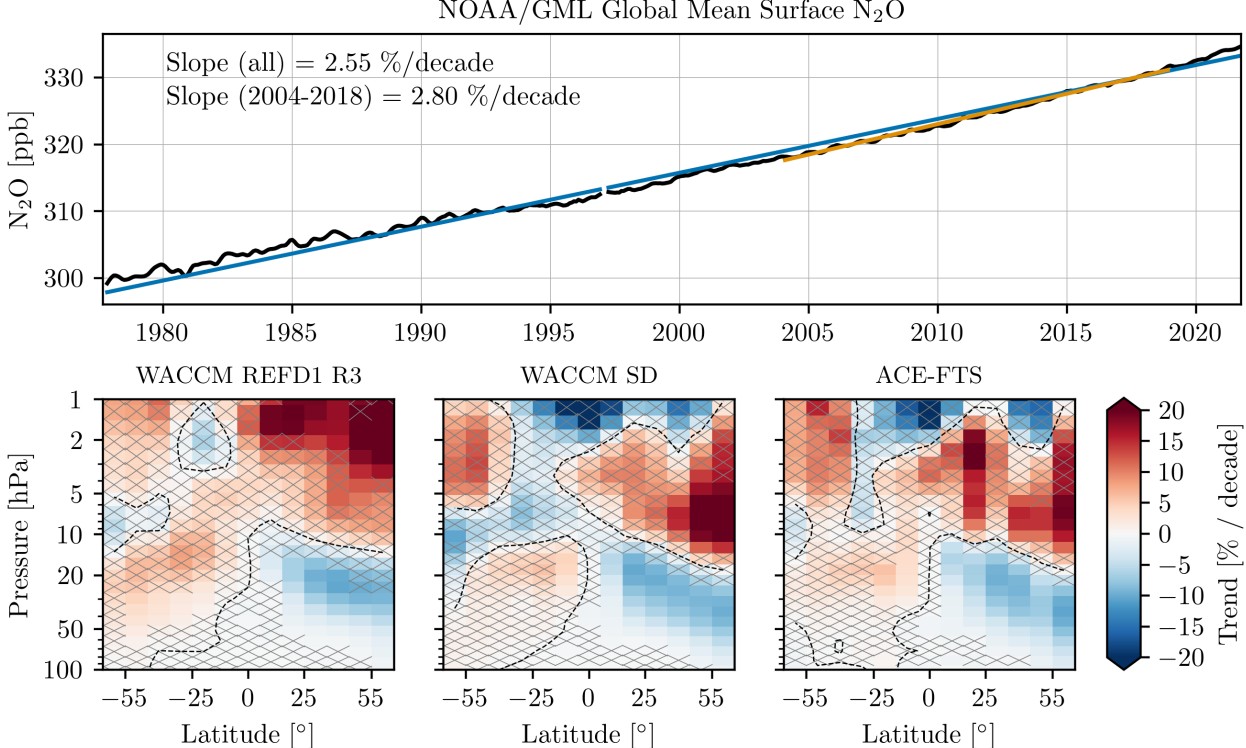

**Figure 2.** Top: Global mean surface $N_2O$ trend. The black line is the observations, the blue line is the trend for the full dataset, and the orange line is the trend for 2004–2018. Bottom: portion of the stratospheric $N_2O$ trend that remains in WACCM and ACE-FTS after the $N_2O$ surface trend has been removed. Trends are for 2004/02 – 2018/12, and calculated with the standard MLR (Equation 2). Hatched regions are insignificant at the $2\sigma$ level. Dashed contours mark the transitions from positive to negative trends.

The $N_2O$ proxy accounts for variability associated with seasonal variations, the QBO, and ENSO, in addition to variability due to a changing BDC. Stolarski et al. (2018) did not consider the effects of the solar cycle and aerosol extinction, even though they have a chemical, rather than dynamical, effect on HCl and ozone. We investigate the effect of including the aerosol and solar cycle proxies in the $N_2O$ MLR in Section 4.

## 4 Results

Trends calculated with the standard MLR and the $N_2O$ MLR are compared to assess the ability of $N_2O$ to control for stratospheric circulation changes. Two different forms of the $N_2O$ MLR are considered, one that uses the ACE-FTS $N_2O$ observations as a proxy, and one that uses the simulated WACCM $N_2O$ as a proxy. The WACCM $N_2O$ predictor is either from the corresponding WACCM run (free-running ensemble member or specified dynamics), or from the specified dynamics WACCM





run when considering observational data. The specified dynamics run was chosen as it most realistically represents the $N_2O$ trend in the ACE-FTS observations (Figures 1 and 2). While the ACE-FTS measurements are perhaps a better representation of the true stratosphere than WACCM, the sparse ACE-FTS sampling pattern could limit the ability to use ACE-FTS $N_2O$

observations as a regression proxy to calculate trends in measurements from other instruments. The ACE-FTS $N_2O$ proxy is also limited by the mission lifetime, whereas WACCM can be run for any time period of interest.

We start by examining the HCl trends for latitudes from 60°S to 60°N and pressure levels from 100 hPa to 1 hPa (Figure 3). This is an update to the results from Stolarski et al. (2018), who focused on a single latitude band, and Bernath et al. (2020), who only discussed the global mean trend. The top row in Figure 3 shows the HCl trends calculated with the standard

MLR. Only one free-running WACCM ensemble member is included due to the similarity between all four members (the other three are provided in the appendix, Figure A2). The WACCM HCl trend is biased low relative to the ACE-FTS observations: the HCl trend depends on surface emission of chlorine-containing gases, so the lower trend in WACCM suggests that the model is assuming a greater reduction in chlorine emissions than has actually occurred. This is likely because the REFD1 model simulations do not include the effect of chlorine containing Very Short Lived Source Gases (VSL-SGs, Plummer et al.,

2021), which have been increasing over the past 20 years (Laube et al., 2022). Hossaini et al. (2019) showed that modelled stratospheric HCl trends are more negative than HCl trends from ACE-FTS when VSL-SGs are not included in the model. As with $N_2O$, there is a distinct hemispherical asymmetry in the standard HCl trends below 10 hPa in both the observations and the model runs. This pattern was first observed by Mahieu et al. (2014), who attributed it to a temporary slowdown of the BDC in the NH relative to the SH.

The middle row of Figure 3 shows the HCl trends calculated with the WACCM $N_2O$ MLR. The largely negative trend is what we expect from decreasing global chlorine emissions. In all cases the $N_2O$ regression produces similar trends in the NH and SH, although the trend remains slightly lower in the SH compared to the NH. These results agree with previous studies showing that changes in stratospheric circulation are responsible for the observed increase in NH HCl over the past two decades (Mahieu et al., 2014; Stolarski et al., 2018), rather than some unaccounted for source of HCl in the NH.

Using ACE-FTS $N_2O$ as the regression proxy works well for ACE-FTS observations, but not for WACCM simulations (bottom row of Figure 3). In the case of the ACE-FTS observations the HCl trend in the NH is greatly reduced when either WACCM $N_2O$ or ACE-FTS $N_2O$ is used as the regression proxy (the HCl trend in the SH increases slightly more when the ACE-FTS $N_2O$ is used in the MLR instead of the WACCM $N_2O$). However, when ACE-FTS $N_2O$ is used as a regression proxy to calculate WACCM HCl trends there is a clear hemispheric asymmetry remaining. This suggests that the ACE-FTS

$N_2O$ observations only work well as a stratospheric circulation proxy for calculating trends in gases measured by ACE-FTS, when there is a perfect sampling match. We also note the the inclusion of aerosol and/or solar cycle proxies in the $N_2O$ MLR has a negligible impact on the HCl trends- this is shown for the ACE-FTS observations in the appendix, Figure A3.

The comparison with the standard HCl trends demonstrates that the $N_2O$ MLR removes most of the positive or insignificant HCl trends by accounting for signals of dynamical variability. Based on this positive result, we next used the $N_2O$ proxy

to determine $O_3$ trends accounting for changes in stratospheric circulation. The top row of Figure 4 shows the $O_3$ trends for 2004/02–2018/12 from the standard MLR (the remaining free-running WACCM ensemble members are shown in the





**Figure 3.** HCl trends for 2004/02 – 2018/12. Top row: trend calculated with standard MLR. Centre row: trends calculated with WACCM $N_2O$ MLR. Bottom row: trends calculated with ACE-FTS $N_2O$ MLR. Hatched regions are insignificant at the $2\sigma$ level. Dashed contours mark the transitions from positive to negative trends.

appendix, Figure A4). The standard trends are broadly consistent between all WACCM runs and the ACE-FTS observations: $O_3$ is increasing above 10 hPa and in the SH, although this increase is less significant in ACE-FTS $O_3$ than in WACCM $O_3$. In all cases there is a tongue of decreasing trend from about 40 hPa to 10 hPa in the NH, with a smaller region of negative trend extending down to 100 hPa. These negative trends in the tropics and NH are a known feature of $O_3$ trends in the 21st century that have been broadly attributed to circulation changes (e.g. Ball et al., 2019, 2020; Bognar et al., 2022; Godin-Beekmann et al., 2022).

The WACCM $N_2O$ MLR successfully accounts for the negative ozone trend in the NH above $\sim$30 hPa: the bins of significant negative trend become insignificant and positive (middle row of Figure 4). At the same time, the WACCM $N_2O$ MLR enhances the region of (largely insignificant) negative ozone trend in the NH and tropics below 30 hPa. The $O_3$ trends for ACE-FTS



O$_3$ Trends - Standard MLR

O$_3$ Trends - WACCM N$_2$O MLR

O$_3$ Trends - ACE N$_2$O MLR

**Figure 4.** O$_3$ trends for 2004/02 – 2018/12. Top row: trend calculated with standard MLR. Centre row: trends calculated with WACCM N$_2$O MLR. Bottom row: trends calculated with ACE-FTS N$_2$O MLR. Hatched regions are insignificant at the 2$\sigma$ level. Dashed contours mark the transitions from positive to negative trends.

are similar whether WACCM or ACE-FTS N$_2$O is used as the regression proxy, however using ACE-FTS N$_2$O as a proxy for calculating trends in WACCM O$_3$ results in trends that are more similar to those from the standard MLR than from the WACCM N$_2$O MLR. As with HCl, the inclusion of a proxy for the solar cycle or for aerosols has minimal effect on the O$_3$ trends from the N$_2$O regression (appendix, Figure A3). Therefore changes in aerosol levels or solar radiation levels are not

significantly impacting stratospheric O$_3$ trends over 2004–2018.

The effect that the N$_2$O proxy has on both O$_3$ and HCl trends can be understood by considering the relationships between the gases. The level of correlation between HCl and N$_2$O is shown in the top row of Figure 5 for the WACCM-SD run. N$_2$O and HCl are anti-correlated throughout the lower and mid-stratosphere because N$_2$O is a long-lived trace gas with a tropospheric source, and HCl is a long-lived trace gas with a stratospheric source. While the anti-correlation is largely due to transport-





driven variability on shorter time-scales, it also implies that long-term changes in transport will have opposite effects on $N_2O$ and HCl. The anti-correlation between HCl and $N_2O$ is thus consistent with their trends from the standard MLR being of opposite sign everywhere below 10 hPa and indicates that the latter are driven by long-term changes in transport. The contour lines in the top left panel of Figure 5 show where the HCl trend is greater than -3%/decade, clearly outlining the region where the standard MLR computes a trend that is inconsistent with tropospheric chlorine trends. In this same region the dynamical

component of the $N_2O$ trend is negative (Figure 2). Fitting the $N_2O$ proxy to remove the portion of the HCl trend caused by transport lowers the resulting HCl trend in the NH, which is now consistent with the chemical signal from the tropospheric chlorine trends. At the same time, it increases the HCl trend in the SH, successfully removing the difference in HCl trends between the hemispheres.

  Turning now to $O_3$, the bottom row of Figure 5 shows that $O_3$ is anti-correlated with $N_2O$ below ∼30 hPa, and correlated

above. Above ∼30 hPa the lifetime of $O_3$ is largely defined by chemical production and loss, while below ∼30 hPa the $O_3$ distribution is controlled by transport. As with $N_2O$ and HCl, the transport-driven anti-correlation between $N_2O$ and $O_3$ below ∼30 hPa suggests that long-term changes in stratospheric circulation will have an opposite effect on $N_2O$ and $O_3$ in this region. Therefore fitting the $N_2O$ proxy to account for the portion of the $O_3$ trend caused by transport lowers the $O_3$ trend in the NH, where the dynamical $N_2O$ trend is negative. Conversely, between ∼10 hPa and 30 hPa in the NH $O_3$ and $N_2O$ are correlated

and the dynamical $N_2O$ trend is negative, so fitting the $N_2O$ proxy to $O_3$ increases the $O_3$ trend (compared to the standard MLR trend). The black contour lines in the bottom row of Figure 5 show where the $O_3$ trend is zero or negative for each version of the MLR, illustrating the increase in the $O_3$ trend in the NH above 30 hPa, and the decrease in the $O_3$ trend in the NH below 30 hPa. Overall, using the $N_2O$ MLR instead of the standard MLR reduces the hemispheric asymmetry in $O_3$ trends above 30 hPa, but increases the asymmetry below 30 hPa. Below 30 hPa changes to the stratospheric circulation induce a positive ozone

trend so there must be some other reason for the negative or insignificant trend that is observed in this region when using the standard MLR.

  We now investigate using the $N_2O$ MLR to calculate trends in OSIRIS $O_3$. Figure 6 shows the OSIRIS $O_3$ trends from the standard MLR, the $N_2O$ MLR with WACCM-SD $N_2O$, and the $N_2O$ MLR with ACE-FTS $N_2O$. The trends are overall very similar to those for WACCM and ACE-FTS $O_3$, with the negative trend from 10-30 hPa in the NH disappearing when the

$N_2O$ MLR is used instead of the standard MLR. $N_2O$ from both ACE-FTS and WACCM-SD work equally well as dynamical proxies for calculating OSIRIS $O_3$ trends. This provides confidence the $N_2O$ observations from ACE-FTS or simulations from WACCM can be used to determine trends in an independent observational dataset.

  The $N_2O$ regression can also be used to study $NO_y$ trends. Surface $N_2O$ emissions are the main source of stratospheric $NO_y$, and both $NO_y$ and $N_2O$ are impacted by BDC changes. By controlling for these effects, the $N_2O$ regression can be

used to determine if the $NO_y$ trend is consistent with increasing $N_2O$ surface emissions, or if there is some other factor affecting the $NO_y$ trend. The $NO_y$ trend from the standard MLR is shown in the top row of Figure 7. The ACE-FTS sunrise and sunset occultations are kept separate to avoid any diurnal effects in the $NO_y$ species. As with $N_2O$ and HCl, there is a distinct hemispheric asymmetry in the $NO_y$ trends from the standard regression. The positive trends in the NH below 10 hPa have similar magnitudes for both WACCM and ACE-FTS, but the negative trend in the SH is lower in ACE-FTS observations



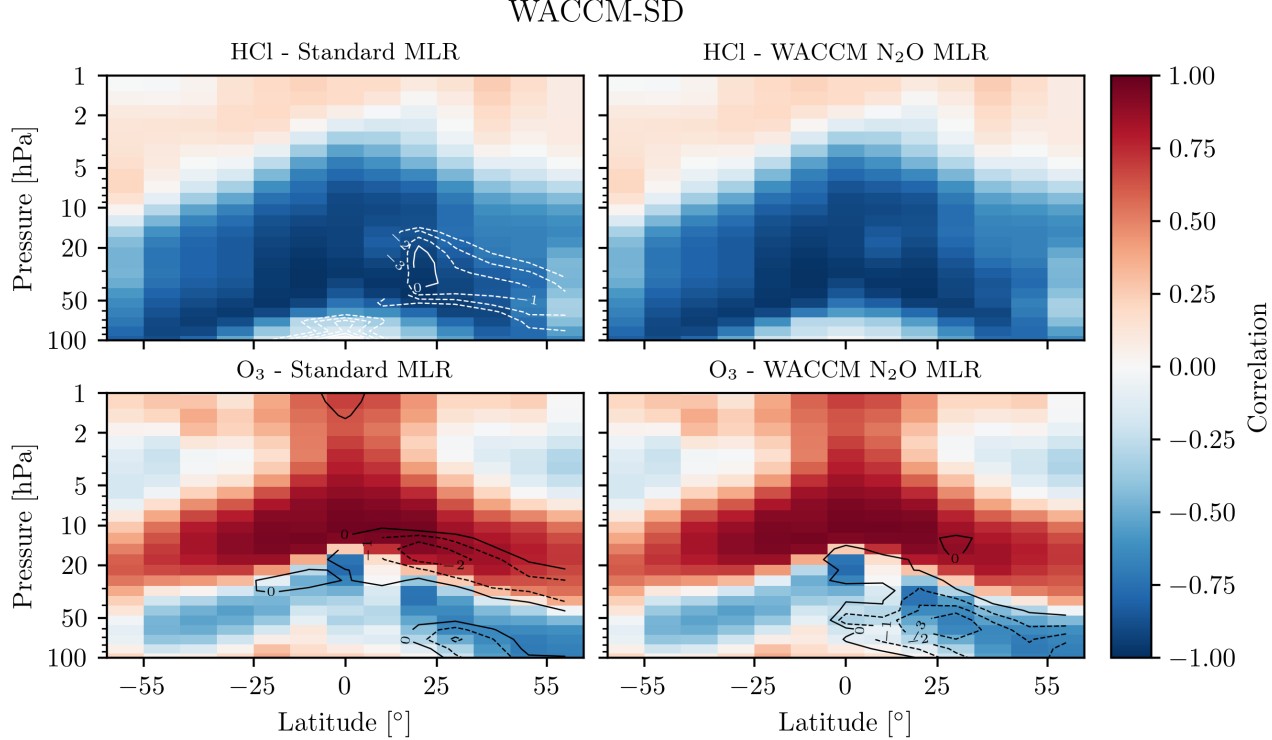

**Figure 5.** Top row: correlation coefficient for WACCM-SD HCl and N$_2$O. White contours show where the HCl trend is greater than -3%/decade. Bottom row: correlation coefficient for WACCM-SD O$_3$ and N$_2$O. Black contours show where the O$_3$ trend is less than 0%/decade. The trend contours in the left row were calculated with the standard MLR, and the trend contours in the right row were calculated with the N$_2$O MLR.

compared to the WACCM results. The WACCM trends look the same if the WACCM NO$_y$ is limited to the five gases used to calculate the ACE-FTS NO$_y$, so this is not the reason for the difference between their trends. It is possible that the differences between WACCM and ACE-FTS NO$_y$ are due to differences in local time as the ACE-FTS NO$_y$ trends agrees better with WACCM at sunset than sunrise, particularly in the NH. The WACCM NO$_y$ results are a daily mean, rather then calculated at a specific local time.

N$_2$O is the main source of NO$_y$, so we expect that using the N$_2$O regression to account for the effect of dynamics will result in an NO$_y$ trend with a comparable magnitude to the N$_2$O emissions trend (2.8%/decade). This does occur in the case of WACCM- both the REFD1 simulations and the specified dynamics run have insignificant trends on the order of 1%-3% per decade, as calculated with the WACCM N$_2$O regression (middle row of Figure 7). However, using the N$_2$O regression (with either WACCM or ACE-FTS N$_2$O) on ACE-FTS NO$_y$ observations results in a largely negative and significant trend. As with

HCl and O$_3$, using ACE-FTS N$_2$O observations as a regression proxy for WACCM NO$_y$ does not adequately account for the trend difference between the hemispheres.





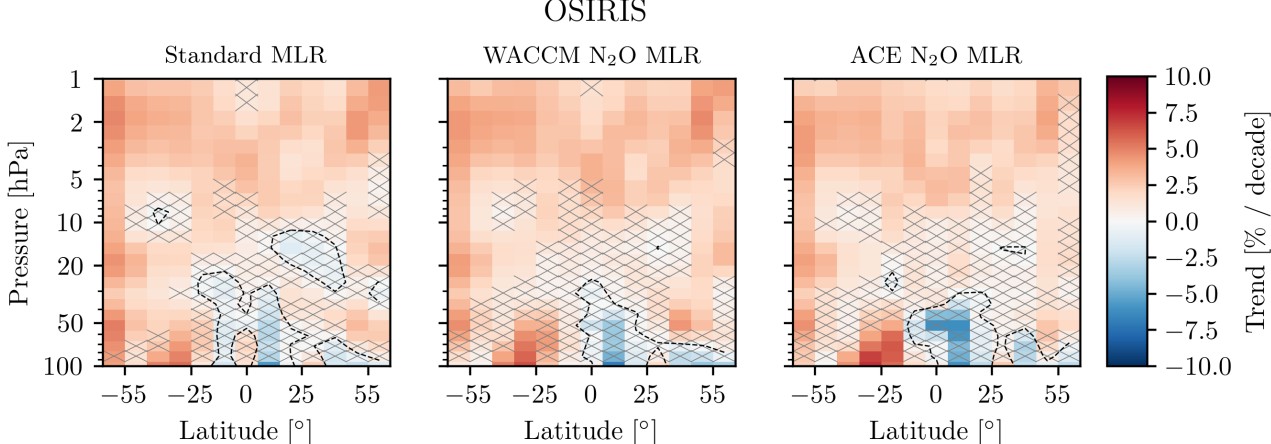

**Figure 6.** OSIRIS $O_3$ trends for 2004/02 – 2018/12. Hatched regions are insignificant at the $2\sigma$ level. Dashed contours mark the transitions from positive to negative trends.

Since $N_2O$ is the main source of $NO_y$, these gases are anti-correlated throughout much of the stratosphere. This means that when using the $N_2O$ proxy, the positive $N_2O$ trend in the SH will increase the $NO_y$ trend in the SH, making it less negative. Similarly, in the NH, the negative trend in the $N_2O$ proxy will make the $NO_y$ trend less positive. For WACCM the

magnitudes of the $N_2O$ and $NO_y$ trends from the standard regression are similar, so we see the expected cancellation in the $N_2O$ regression. For ACE-FTS the $NO_y$ trends in the SH are lower than expected based on the $N_2O$ trend, so the $N_2O$ proxy cannot fully explain the decreasing $NO_y$. However, the overall changes in the trends when using the $N_2O$ MLR instead of the standard MLR are consistent with what we expect from the relationship between $NO_y$ and $N_2O$. Further understanding the differences between ACE-FTS and WACCM $NO_y$ trends requires a more detailed study.

**5    Conclusions**

Several recent studies showed that stratospheric ozone has declined in the tropics and NH throughout the 21st century, despite the overall success of the Montreal protocol in reducing ozone depletion in the SH and upper stratosphere (e.g. Ball et al., 2018, 2019; SPARC/IO3C/GAW, 2019; Bognar et al., 2022; Godin-Beekmann et al., 2022). The remaining negative ozone trend is thought to be caused by changes in the BDC that result in slower moving air in the NH relative to the SH (e.g.,

Ploeger and Garny, 2022; Strahan et al., 2020). The present work uses $N_2O$ observations from ACE-FTS and simulations from WACCM to account for BDC changes in the MLR used to calculate ozone trends.

Ozone trends from 2004–2018 are consistent between observations from OSIRIS and ACE-FTS. Trends in $O_3$ simulations from both free-running and specified dynamics WACCM runs also agree very well with the observational trends. In each case, the standard MLR results show an $O_3$ decrease in the NH between about 10 hPa and 30 hPa, as well as $O_3$ decrease at lower

levels. By using the $N_2O$ MLR instead of the standard MLR, the negative $O_3$ trend in the NH above 30 hPa is eliminated







**Figure 7.** $NO_y$ trends for 2004/02 – 2018/12. Top row: trend calculated with standard MLR. Centre row: trends calculated with WACCM $N_2O$ MLR. Bottom row: trends calculated with ACE-FTS $N_2O$ MLR. Hatched regions are insignificant at the $2\sigma$ level. Dashed contours mark the transitions from positive to negative trends.

or becomes insignificant. However, the $N_2O$ proxy cannot explain the negative ozone trends that are observed in the tropics below 20 hPa. $N_2O$ time series from both WACCM and ACE-FTS work well as a regression proxy for $O_3$ from OSIRIS, but the ACE-FTS $N_2O$ MLR does not improve upon the results of the standard MLR when calculating WACCM $O_3$ trends. The results of using the $N_2O$ MLR to determine stratospheric ozone trends suggests that the observed $O_3$ decrease in the NH above 30 hPa is caused by changes in the BDC, but that the negative $O_3$ trend at lower levels is not. These results agree with those from Weber et al. (2022), who found that including proxies for dynamical variability in the standard MLR model increased column $O_3$ trends in the NH, but did not change column $O_3$ trends in the tropics. It is possible that the negative $O_3$ trend below 30 hPa is due to changes in the tropopause height, rather than in upwelling. Bognar et al. (2022) found that using tropopause relative coordinates reduced the significance and magnitude of the negative $O_3$ trends over 2000–2021 in the tropics, up to 7



km above the tropopause. The $N_2O$ proxy cannot account for changes in the tropopause height because $N_2O$ is relatively inert in both the upper troposphere and lower stratosphere (ie. Figure 2, the $N_2O$ trend in the tropical stratosphere below 50 hPa is the same as the surface $N_2O$ trend).

$N_2O$ from WACCM successfully accounts for the hemispherical asymmetry that is present in HCl trends computed using the standard MLR. $N_2O$ from ACE-FTS can also be used as a regression proxy, but it works best when considering other

ACE-FTS observations, rather than WACCM results. The ability of the $N_2O$ proxies to explain the HCl increase in the NH implies that this increase is not caused by rising chlorine emissions, but rather by changes in transport. This is consistent with observations showing that chlorine emissions have largely been declining (Laube et al., 2022).

Lastly, we showed that trends in $NO_y$ determined using the standards MLR have a strong asymmetry, with negative trends in the SH and positive trends in the NH. This is true for both ACE-FTS observations and in WACCM simulations. When using

the $N_2O$ MLR to find the $NO_y$ trends, the WACCM $NO_y$ trends become mostly insignificant and have a similar magnitude as the $N_2O$ surface emissions. This is what is expected as the main $NO_y$ source is $N_2O$. This is not the case for the ACE-FTS $NO_y$ trends, which are significantly negative in most bins when calculated with the $N_2O$ MLR. Despite this, the ACE-FTS $NO_y$ trends from the $N_2O$ MLR are consistent with what is expected from the relationship between $N_2O$ and $NO_y$, given the trends from the standard MLR.

*Code and data availability.* OSIRIS data are available at https://research-groups.usask.ca/osiris/data-products.php#OSIRISLevel2DataProducts (Zawada et al., 2022).

The WACCM results are available at ftp://odin-osiris.usask.ca/Models.

Instructions for downloading the OSIRIS and WACCM files are at https:// research-groups.usask.ca/osiris/data-products.php#Download

ACE-FTS data are available at https://databace.scisat.ca/level2/ (ACE-FTS, 2022).

ACE-FTS data quality flags are available at https://doi.org/10.5683/ SP2/BC4ATC (Sheese and Walker, 2022).

The NOAA/GML HATS $N_2O$ is available at https://gml.noaa.gov/hats/combined/N2O.html (NOAA/GML, 2022).

The LOTUS regression code and documentation is available at https://arg.usask.ca/docs/LOTUS_regression/index.html (Damadeo et al., 2022).

**Appendix A:  Extra Figures**

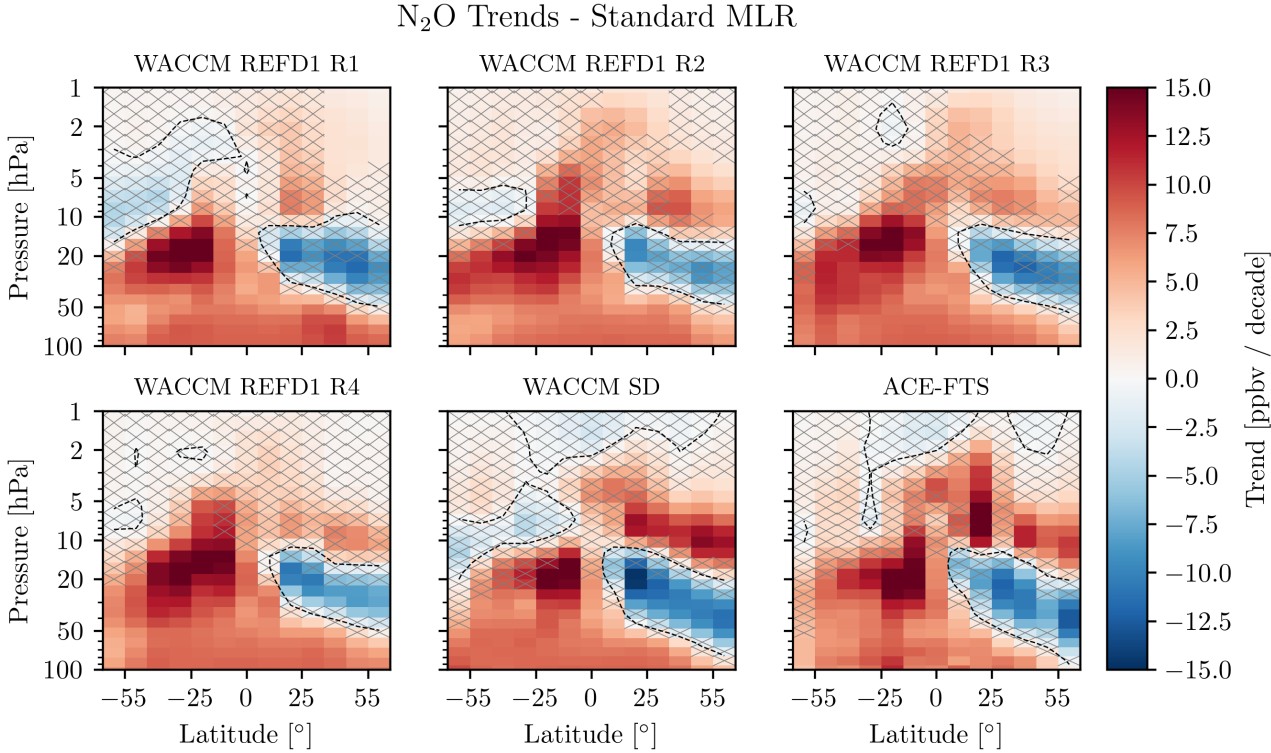

**Figure A1.** N$_2$O trend in ACE-FTS and WACCM trends in units of ppbv/decade for 2004/02 – 2018/12, as calculated with the standard MLR (Equation 2). Hatched regions are insignificant at the 2$\sigma$ level. Dashed contours mark the transitions from positive to negative trends.

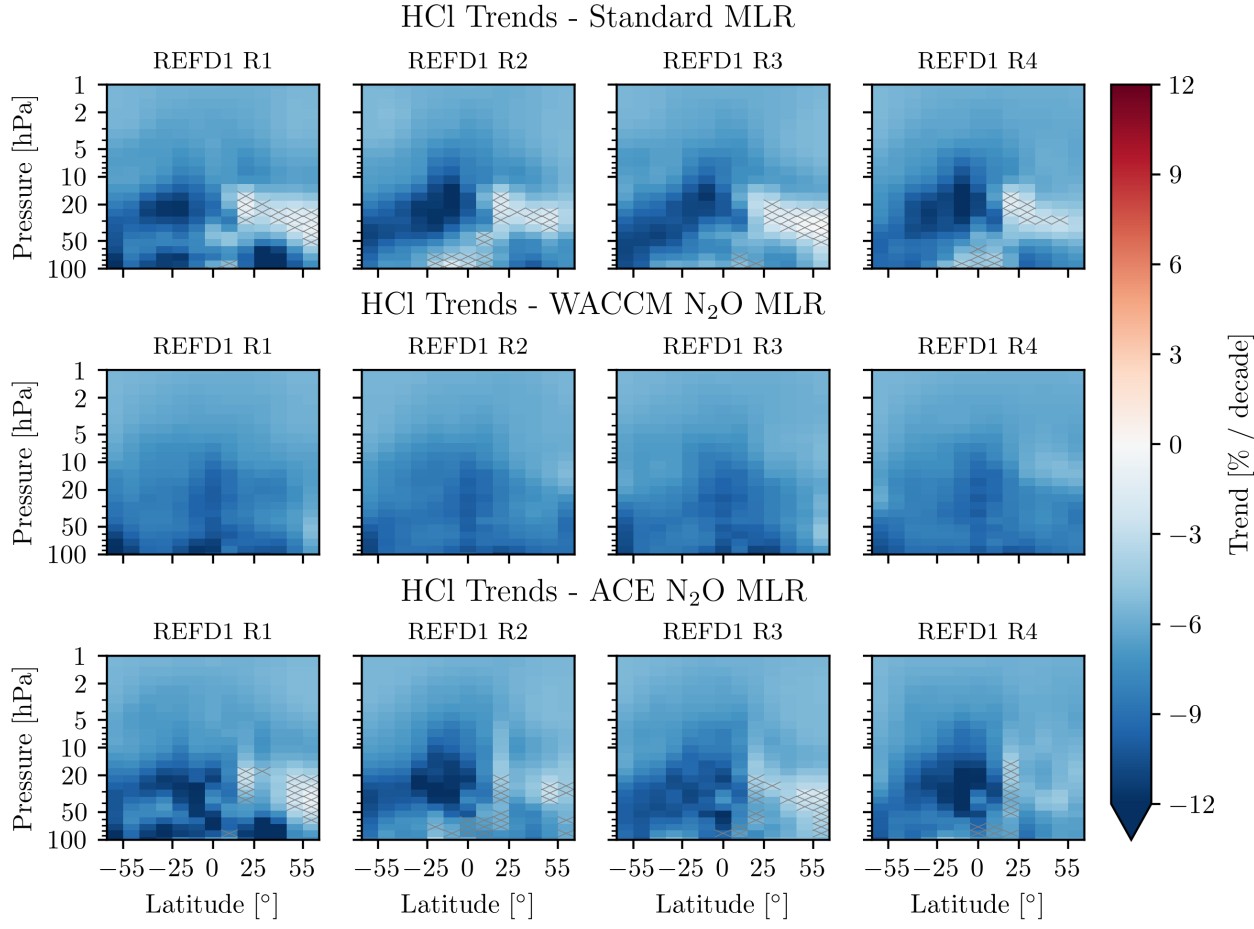

**Figure A2.** WACCM REFD1 HCl trends for 2004/02 – 2018/12. Top row: trend calculated with standard MLR. Centre row: trends calculated with WACCM $N_2O$ MLR. Bottom row: trends calculated with ACE-FTS $N_2O$ MLR. Hatched regions are insignificant at the $2\sigma$ level.



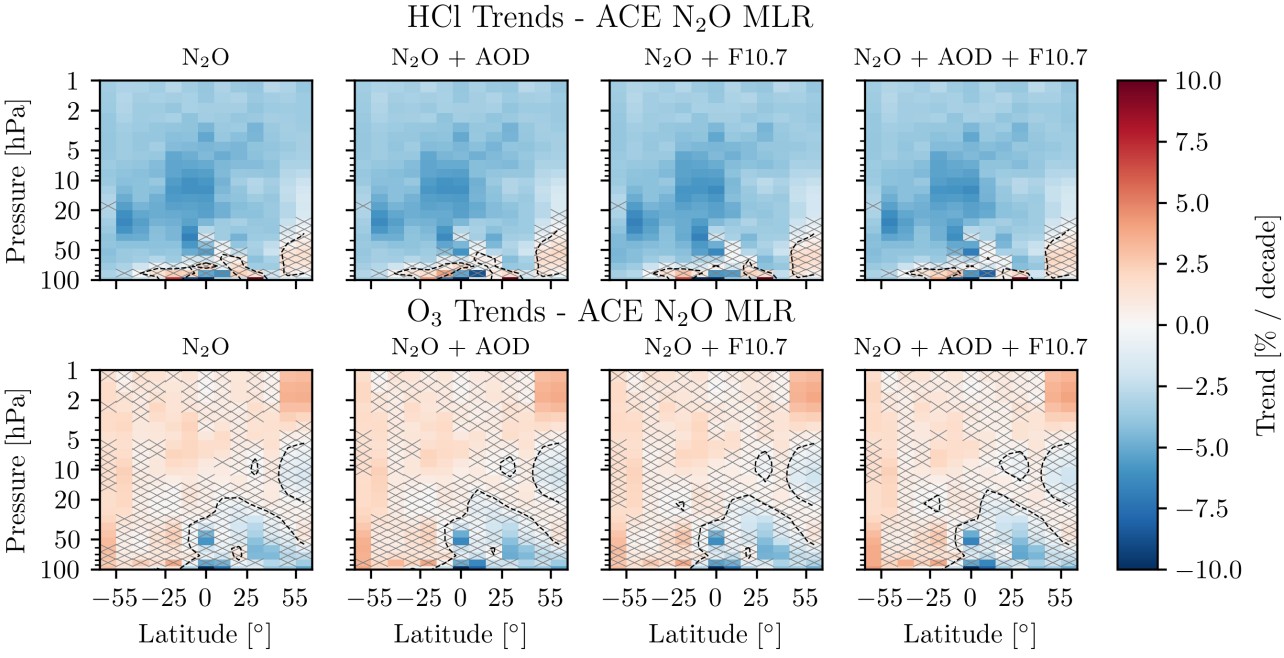

**Figure A3.** ACE-FTS trends for 2004/02 – 2018/12. Top row: HCl trends calculated with ACE $N_2O$ MLR. Bottom row: $O_3$ trends calculated with ACE-FTS $N_2O$ MLR. Hatched regions are insignificant at the $2\sigma$ level.

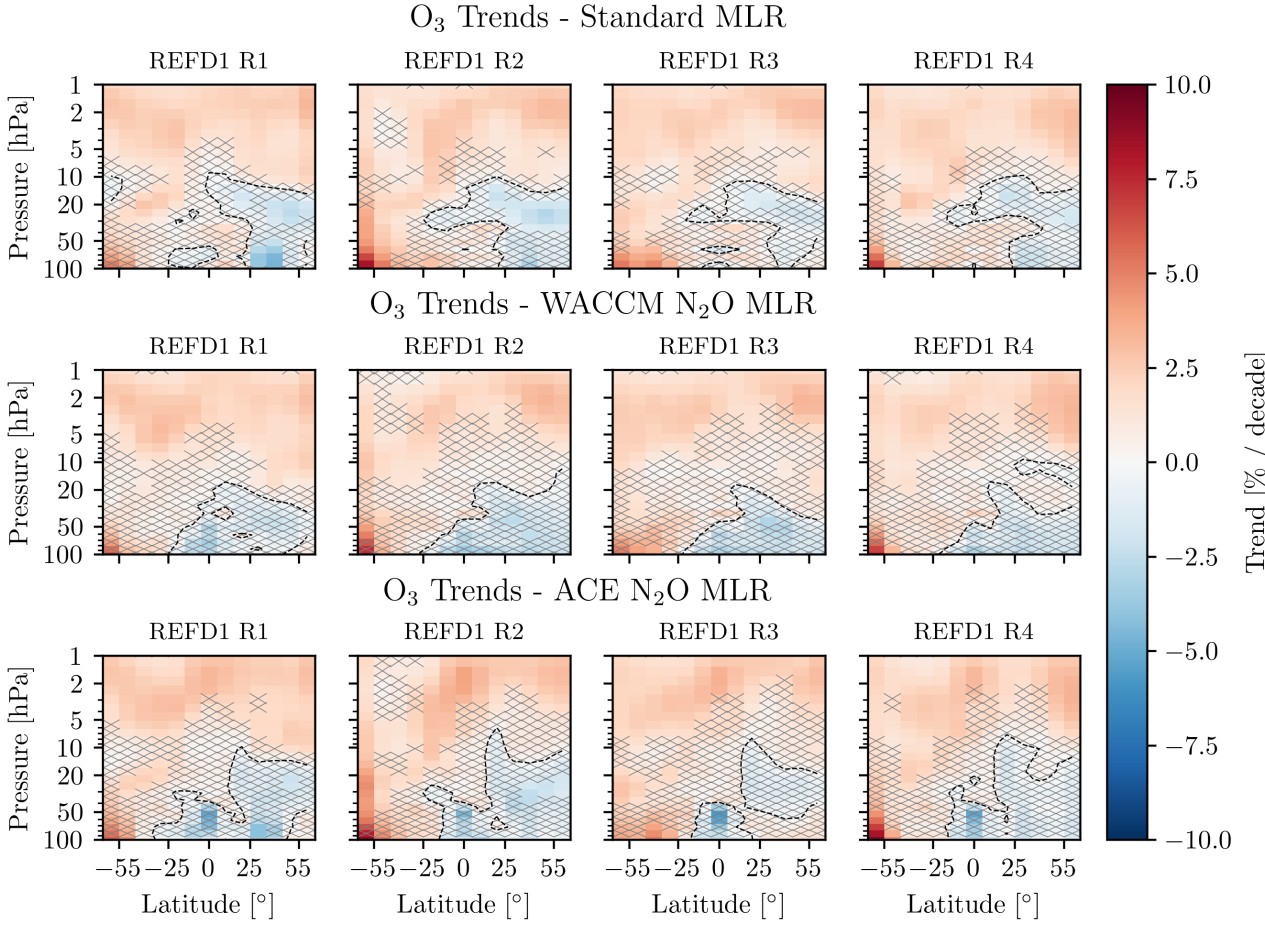

**Figure A4.** WACCM REFD1 $O_3$ trends for 2004/02 – 2018/12. Top row: trend calculated with standard MLR. Centre row: trends calculated with WACCM $N_2O$ MLR. Bottom row: trends calculated with ACE-FTS $N_2O$ MLR. Hatched regions are insignificant at the $2\sigma$ level.



*Author contributions.* KD performed the analysis and prepared the manuscript. ST, AB, DZ, and DD provided input on the method and analysis. PS and KW provided guidance on using the ACE-FTS data. WR provided the WACCM results. All authors provided significant feedback on the manuscript.

*Competing interests.* We declare that none of the authors have any competing interests.

*Acknowledgements.* This research has been supported by the Canadian Space Agency (grant no. 21SUASULSO). The authors thank the
Swedish National Space Agency and the Canadian Space Agency for the continued operation and support of Odin-OSIRIS. The Atmospheric Chemistry Experiment (ACE) is a Canadian-led mission mainly supported by the CSA and the NSERC, and Peter Bernath is the principal investigator. The National Center for Atmospheric Research is sponsored by the U.S. National Science Foundation. WJR was also supported as part of the Aura Science Team under NASA Grant 80NSSC20K0928.



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
