# Peer review of "$N_2O$ as a regression proxy for dynamical variability in stratospheric trace gas trends"

_EGUsphere, 2023_

## Author Comment (AC1)

Response to reviews: N2O as a regression proxy for dynamical variability in stratospheric trace gas trends

Thank you for the helpful comments! Responses to each question/comment are included here in blue text.

RC1

1.  Including the goodness of fit (R2) distribution for both regression models, specifically for ACE-FTS and WACCM-SD, would provide valuable insights into the reliability of trend estimates and the quality of each regression model. This would enhance the assessment of the trend estimates obtained using the LOTUS setup.

    The R2 values for WACCM and ACE HCl and O3 trends were added to the appendix, and the results are discussed in Section 4 of the manuscript.

2.  Recently, Li et al. (2023, ACPD) employed a multivariate regression model with the eddy heat flux as a dynamical proxy to demonstrate inter-hemispheric asymmetry in stratospheric ozone trends. It would be interesting if the authors could compare N2O as a dynamical proxy to the eddy heat flux to determine if N2O proxy performs better.

    Yes, this would be very interesting to look at. Typically, the EHF proxy is based on the EHF at 100 hPa from 45-75 degrees in each hemisphere (eg. Weber et al. 2011, 2022), so we do not expect that it would work as well as N2O for explaining the dynamic variability in trace gases at lower latitudes. Indeed, Weber et al. (2022) mention that including an EHF proxy did not change the tropical column ozone trend. It would certainly be valuable to look at the effects of an EHF proxy that varies in altitude/altitude. However, doing so would require a significant amount of work (repeating the whole analysis for a new scenario) and will not change any of the conclusions made in the manuscript about the N2O proxy so we believe it is outside the scope of this paper.

    Unfortunately, we cannot compare to Li et al. 2023 as the preprint available online does not discuss the results of using an EHF proxy (there is mention of an EP flux proxy). A discussion of the EHF proxy results from Weber et al. (2022) and from the SPARC LOTUS report is provided on lines 67-72.

    Weber, M., Dikty, S., Burrows, J. P., Garny, H., Dameris, M., Kubin, A., Abalichin, J., and Langematz, U.: The Brewer-Dobson circulation and total ozone from seasonal to decadal time scales, Atmos. Chem. Phys., 11, 11221–11235, https://doi.org/10.5194/acp-11-11221-2011, 2011.

    Weber, M., Arosio, C., Coldewey-Egbers, M., Fioletov, V. E., Frith, S. M., Wild, J. D., Tourpali, K., Burrows, J. P., and Loyola, D.: Global total ozone recovery trends attributed to ozone-depleting substance (ODS) changes derived from five merged ozone datasets, Atmos. Chem. Phys., 22, 6843–6859, https://doi.org/10.5194/acp-22-6843-2022, 2022.

3. To improve the clarity of the presentation, it would be beneficial to compare the evolution of trace gases at selected latitude/altitude bins (e.g., 30S, 30N, 30 hPa, 5 hPa) as well as the regression fits. This approach would facilitate the visualization of differences between ACE-FTS and WACCM simulated HCl, N2O, O3, and NOy, making it easier to compare the model/observations as well as performance of the regression models.

Figures compaing the regression fit to the time series for WACCM-SD and ACE-FTS HCl, O3, and NOy have been added to the appendix.

4. The authors argue that the discrepancy in HCl trend between WACCM and ACE-FTS results from the absence of Cl-containing VSLS species in the WACCM simulations. It would strengthen their claim if they could provide additional evidence, such as comparing the vertical distribution of HCl between WACCM and ACE for the initial and last five years of the time series.

This is certainly something that is worth looking into further. However, the purpose of our paper is only to demonstrate the use of N2O as a proxy for long-term circulation changes in trace gas trend calculations. Therefore, we feel it is outside the scope to determine the exact cause of the different HCl trends in WACCM simulations and ACE-FTS observations. Statements saying that the difference being caused by VSLS is only a theory and that further work on this topic is needed have been added to the manuscript.

Minor comment:

In line 145, the phrase "In the middle row" is unnecessary as the figure caption already provides this information.

In line 146, the word "emission" should be replaced with "emissions of chlorine-containing source gases" or something similar.

I think this is referring to lines 195 and 196. Has been fixed.

RC2

Minor comments:

1. I'm somewhat confused about the motivation of the study to "isolate trends due to circulation changes from trends due to ozone depleting substances", as stated e.g. in the abstract (P1, L4). How can the total effect of ODS be separated, as it includes also an effect via ozone-induced circulation changes. Isn't the N2O just a proxy for stratospheric circulation changes, regardless of their causes (e.g. ODS). Also, I don't see this as a problem for the paper, just the reasoning should be clarified, here and throughout the paper.

You are correct, this wording has been modified. The N2O is indeed just a proxy for all circulation changes.

2. I understand that the standard MLR model assumes an instantaneous response, as it includes no time lags (P5, L27). Especially for ENSO this could be problematic, as the stratospheric response to SST changes is likely delayed in the stratosphere. Including lag time in the ENSO regressor

could improve the MLR model for stratospheric circulation, as shown e.g. by Diallo et al. (2019, ACP, 10.5194/acp-19-425-2019) and could be mentioned.

We looked at the effect of including a lag in the ENSO term of 1 month to 1 year and found that it had a negligible effect on the calculated ozone trends. This is now mentioned in Section 3 of the manuscript.

3.  I'm somewhat unsure about the conclusion that "the N2O proxy cannot explain the negative ozone trends that are observed in the tropics below 20hPa" (e.g. P14, L296, or similarly in the abstract). First, I don't see negative O3 trends in the tropics below 20hPa in this study (at least not significant, Fig. 4). Second, including the N2O proxy in the MLR causes the O3 trends in this region to be more negative. So just based on the data shown here, I'd conclude that there is a slow-down in tropical upwelling during 2004-2018 which causes positive tropical O3 trends. This is not what other studies showed, although for different periods (e.g. Ball et al., 2018 for 1998-2016). It is also counter-intuitive to the O3-response to an expected increase in tropical upwelling over time, what climate models predict on the long term. I'd find it interesting to see the trend in tropical upwelling w* (e.g. as contours in Fig. 4), to see how this changes in the WACCM simulation over the considered period. I guess the differences to previous studies are explainable by the different periods considered. Anyway, I suggest to be more careful with the discussion and relation to past studies.

First point: Thank you for pointing this out. We meant to refer to the negative (largely insignificant) ozone trend below 30 hPa in the Northern hemisphere. This has been corrected.

Second point:

Given that the O3 trends in the tropics below 20 hPa are not significant when using the N2O proxy, it is not possible to draw conclusions about changes in tropical upwelling based on our O3 trend results. We are mainly focused on an increase in transport with the Brewer-Dobson circulation in the SH relative to the NH. As the tropical region is the transition region between positive and negative trends it is trickier to decipher. The N2O trends with changes in surface emissions removed (Fig. 2) suggest that there might be increased upwelling in the SH tropics and decreased upwelling in the NH tropics below 20 hPa. This seems to be roughly consistent with the transport induced component of the O3 trend being positive in the NH tropics (as here the O3 trend becomes more negative when the N2O proxy is applied). We decided to not include a detailed discussion of these results as the tropical ozone trend are not significant in most cases.

We decided to not include WACCM w* in our figures as transport changes in regions other than the tropical pipe are not only driven by w* but also by meridional transport and mixing. It is certainly true that the specific time period we are considering interferes with the ability to compare closely with other studies. As our results are for the relatively short time frame of 2004-2018 they do not necessarily contradict the expected increase in upwelling over a long period of time.

Specific comments:

P2, L26: A clear relation of the hemispherically asymmetric trace gas trends to circulation changes was shown by Stiller et al. (2017, ACP, https://doi.org/10.5194/acp-17-11177-2017), involving a latitudinal circulation shift. This could be mentioned here as well. This is now mentioned

P5, L142: I'd add "... N2O, below about 10hPa", as above the signs of the trends could be opposite (Fig. 1). Done

P6, L162: Better say "small" instead of "minimal"? It's a max 1%/dec trend due to photolysis changes compared to ~5%/dec net trend values. Done

P8, L209: I'd delete "positive". Done

P11, L251: The result here that negative O3 trends below about 30hPa are not explained by circulation and transport is different from what other recent papers proposed (e.g., Wargan et al., 2018; Orbe et al., 2020). I think this would be worth a more thorough discussion. This is now mentioned in the conclusion of the manuscript.

P13, L288: more precisely "...result in air moving slower and deeper through the stratosphere in the NH ..." Done

Technical corrections:

P2, L27: year missing for Abalos et al. Done

P4, L101: time series Done

P8, L206: delete "the" Done

P12, L67: agree Done

P15, L313: standard Done